# Prostate Cancer, Treatment and Response of the Hematological System in Mexican Population

**Shaila Cejudo-Arteaga** [1] **, Marco Antonio Ramírez-Reyes** [2]**, Marco Antonio Badillo-Santoyo** [3]**,
Erika Martínez-Cordero** [4]**, Felipe Farías-Serratos** [4] **and María Maldonado-Vega** [4,*]

[1]  Facultad de Medicina, Benemérita Universidad Autónoma de Puebla, 4 Sur #104, Colonia Centro, Puebla de Zaragoza 72000, Mexico; shace06@gmail.com
[2]  Radiation Oncology Service, Bajio Regional High Specialty Hospital, Blvd. Milenio #130, Colonia San Carlos La Roncha, León, Guanajuato 37544, Mexico; ramirezrmarco@gmail.com
[3]  Oncologic Urology Service, Bajio Regional High Specialty Hospital, Blvd. Milenio #130, Colonia San Carlos La Roncha, León, Guanajuato 37544, Mexico; marcoabs2@hotmail.com
[4]  Planning, Teaching and Research Department, Bajio Regional High Specialty Hospital, Blvd. Milenio #130, Colonia San Carlos La Roncha, León, Guanajuato 37544, Mexico; ericordero@hotmail.com (E.M.-C.); felipe_serratos@yahoo.com (F.F.-S.)
*  Correspondence: vega.maldonado.m@gmail.com

**Abstract:** Androgen deprivation therapy (ADT) is the basis for the control of prostate cancer. High levels of prostate-specific antigen (PSA) and high Gleason grade correlate, define the aggressiveness of the cancer in order to establish its treatment and prognosis. This work evaluated the response of 910 patients diagnosed with prostate cancer, separated into three groups according to their response to treatment by ADT: (1) sensitive (TSPC); (2) palliative and did not accept treatment, and (3) group with recurrence or treatment resistance (TRPC). All patients with prostate cancer treated with ADT, and regardless of whether or not they had undergone surgery or taken to radiotherapy, presented with anemia. The hematological response due to the leukocyte/lymphocyte index (L/L) is increased at the end of treatment, possibly due to inflammatory processes generated by cancer, and baseline overweight and obesity. Patients with biochemical relapse exhibit a higher platelet count, suggesting that these cells could participate in the recurrence process and in metastasis (78%) in these patients. The coagulation index (INR) could be an indicator of the platelet response to be considered during the treatment and monitoring of patients.

**Keywords:** androgen deprivation therapy; anemia; leukocytes; platelets; prostate cancer

## 1. Introduction

Prostate cancer was ranked second in the world in 2020 with 1,414,259 new cases; Mexico registered 26,742 cases [1,2]. The risk factors for prostate cancer include age > 65 years, black race, first-line family history, genetic alteration [3], androgen receptor genes KLK2 and KLK3 [4] and prostate-specific antigen (PSA) levels [5–7]. Co-morbidity due to obesity and smoking are related to an increased risk for and aggressiveness of prostate cancer [8,9]. The inflammatory processes such as prostatitis and immunological response by T cell lymphocytes participate in the evolution of prostate cancer [10].

Androgen deprivation therapy (ADT) of testosterone is basic in the treatment of prostate cancer [11,12]. In mammals, sex hormones are regulated by the hypothalamic pituitarygonadal axis (HPG) and start in the hypothalamus, which secretes the gonadotropic hormone (GnRH) that reaches the portal circulation of the pituitary gland where it binds to its receptors and there is the subsequent secretion of hormones including Luteinizing (LH) and Follicle-Stimulating (FSH) [11]. In men, LH acts on the testes stimulating the Leyding cells that produce the testosterone [13,14] directly related to the secretion of PSA. Regulation of testosterone production from HPG is achieved by negative feedback from the testosterone, while androgen receptors that bind testosterone are regulated

by the 5α-dihydrotestosterone (DHT) produced in the prostate, adrenal glands, which is maintained during the inflammatory processes of the prostate gland as a process of atrophy [15–18]. In prostate cancer, some patients demonstrate disease progress with elevated PSA, increased testosterone and metastases [3,6,19,20]. The accumulation of dehydroepiandrosterone (DHEP) escapes to the ADT, generating dihydrotestosterone (DHT) as a potent androgen for maintaining testosterone production. Antagonists of the androgen synthesis pathways from the HPG axis have been used to limit the production of DHT, thus metabolic pathways are activated in patients with recurrence [21–23]. Grade 5 Gleason refers to high risk; prostate cancer staging is complemented by the TNM system [7,10,24,25]. these aspects are relevant for identifying each patient, predicting response to treatment and the prognosis of the disease.

ADT in prostate cancer reduces testosterone below 5% of the normal value and serum estradiol <30% generates severe hypogonadism [26], bone mineral loss reducing its density and causing fractures; in other cases, ADT induces weight gain, insulin resistance, and diabetes. Among the sexual changes we find erectile dysfunction, gynecomastia, decreased testicular and penile size, changes in libido, hair loss, neuropsychological alterations, and reduced Quality of Life (QoL) due to fatigue and persistent anemia [3,27,28]. Treatments are trans-urethral resection of the prostate (TURP) and ADT can work as a single therapy or be combined with surgical processes depending on the individual response [29] such as simple or bilateral scrotal orchiectomy (OSB), and laparoscopic radical prostatectomy (LRP). Localized radiotherapy can be applied to individuals with low-risk prostate cancer, or else, in palliative cases due to metastasis [30].

Preclinical studies reveal an influence of androgens on the maturation of neutrophils, lymphocytes and platelets, but the effects during ADT are not known. The presence of androgen receptors has been demonstrated in platelets [31,32], these are suspected to play an important role in the recurrence of prostate cancer and metastatic processes [6]. What are the clinical differences observed in patients with prostate cancer at the end of ADT treatment? This work describes a population with prostate cancer at the baseline condition before starting treatment and at the end (ADT, surgery, radiotherapy and chemotherapy) evaluating the effects on the hematological system.

## 2. Materials and Methods

### 2.1. Data

Based on clinical reporting and the patients' Klinic$^{MR}$ follow-up system at our hospital, data and information were captured during the period 2007–2021 of patients diagnosed with prostate cancer. The following data corresponded to the moment of diagnosis and to the end of treatment: age, PSA concentration; Gleason score; Gleason grade 1 to 5; cancer treatment; chronic diseases; metastases, comorbidities; recurrence, weight; height, and habits (smoking, drinking).

### 2.2. Study Design

This was a retrospective, observational, descriptive study for Mexican males diagnosed with prostate cancer. The protocol was authorized by the Research Ethics Committee with registration number CEI-22-2021.

### 2.3. Population

Patients diagnosed with prostate cancer are residents of the Bajio Region, which includes five states in the Center of the Mexican Republic (Guanajuato, Michoacan, San Luis Potosi, Queretaro and Zacatecas). The predominant ethnic groups in these regions were Chichimecas, Purepechas, Jonaces, Otomi and Nahuatl, which over time mixed with the Spanish and the miscegenation that is currently observed is presented.

The study included 910 patients with prostate cancer. All were diagnosed and treated at the Oncology Urology, Medical Oncology, and Surgical Oncology services. The population was organized into prostate-cancer three groups as follows:

1.  sensitive to treatment(s) (TSPC);
2.  admitted in a palliative state and with abandonment of treatment (AT), and
3.  resistance to treatment(s) or presented biochemical recurrence (TRPC).

*2.4. Clinical Test*

Data on blood biometry and counts of the different blood elements prior to treatment and at treatment end were considered in the analysis of the patients' condition; the index of lymphocytes/leukocytes, and platelets were calculated. Another element considered was the INR. Clinical tests: blood chemistry, creatinine, cholesterol, hemoglobin, leukocytes, lymphocytes and platelets.

*2.5. Prostate Cancer Surgical Procedures*

Patients were organized by the type of drug treatment and the surgical procedure that they underwent: as follows: ADT (androgen deprivation therapy); TURP (Trans-Urethral Resection of the Prostate); OSB (Bilateral Scrotal Orchiectomy); RRP (Radical Retropubic Prostatectomy) and Radiotherapy (RT).

*2.6. Statistical Analysis*

The data were managed by study groups, or by means of separation by age by surgical procedures and radiotherapy (RT). Statistical analysis was performed using SPSS statistical software. Results were expressed as mean, median, and standard deviation (SD). A comparison between groups was performed using a two-tailed Student's *t*-test and the comparison of variance among multiple groups was performed using a one-way analysis of variance (ANOVA) with the Tukey multiple comparison test or the Dunnett multiple comparison test. $p < 0.05$ was considered statistically significant.

## 3. Results

### 3.1. Characteristics of the Population with Prostate Cancer

In Table 1 the population studied has a median age of 69 years, with only 2% of patients under 50 years of age. The registration of the Body Mass Index (BMI) in the population indicated that 49% are overweight and obese, in contrast with only 27% with normal BMI and 2% being underweight; we had no information on the remaining cases. Clinical data showed that serum hemoglobin had a significant decrease between baseline vs. end of treatment of two units (g/dL), although critical cases of anemia were also reported with mean values of 11.6 g/dL vs. 10.9 g/dL, in 37% of the patients at the start of treatment vs. 47% of patients at the end of treatment. The leukocyte/lymphocyte index increased at the end of treatment 1.4 times. Platelets did not exhibit a significant change in their count between the beginning and the end of treatment; however, there were cases of thrombocytopenia with an average count of $91–10^3$ cells/μL in 4.2% of patients at the beginning of treatment vs. $93–10^3$ cells/μL in 9.3% of patients at the end of treatment. Thrombocytopenia was also present during the treatment with $479–10^3$ cells/μL (2.5%) at baseline and $471–10^3$ cells/μL (3%) at the end, very similar to the changes in INR. The data for creatinine and cholesterol did not show significant changes in the median of the population; however, the extreme cases are presented in Table 1. It is important to clarify that the oncological clinical treatment of prostate cancer has an average duration of 3 and up to 5 years; in cases of recurrence and treatment resistance, complementary surgical alternatives are applied with longer monitoring. Comorbidities are presented in Table 1, finding 24% of patients with *Diabetes mellitus* (DM) and 41% with arterial hypertension (HAS) coupled with smoking and alcoholism habits in 57% and 56%, respectively. Urinary tract infections (UTI) at baseline were 29% and 8% of renal diseases were related to creatinine changes. The interaction between factors of prostate cancer and comorbidities in these patients impacts with a mortality of 32%.

**Table 1.** Clinical characteristics and constitution of the population of patients treated for prostate cancer.

| Characteristics | Median (Percentile 25–75) | Range |
|---|:---:|:---:|
| Age (years) | 69 (63–75) | 43–95 |
| BMI: *n* (%) <br> normal weight <br> underweight <br> overweight <br> obesity | 241 (27) <br> 20 (2) <br> 289 (32) <br> 157 (17) | |
| Hemoglobin baseline (g/dL) | 14 (12–15) | 3.5–20 |
| Hemoglobin after (gr/dL) | 12.4 (10–14) * | 2.5–20 |
| Index L/L baseline | 4.94 (1.3–50) | 1.3–50 |
| Index L/L after | 6.72 (0.86–50) * | 0.89–50 |
| Platelets baseline (cel/uL) | 236 ± 126 | 9–719 |
| Platelets after (cel/uL) | 237 ± 127 | 9–719 |
| INR baseline (second) | 1.06 ± 0.63 | 0.75–10.93 |
| INR after (second) | 1.08 ± 0.56 | 0.76–4.19 |
| Creatinine baseline (mg/dL) | 0.9 (0.8–1.10) | 0.3–46 |
| Creatinine after (mg/dL) | 0.9 (0.8–1.30) | 0.2–13 |
| Total, cholesterol baseline (mg/dL) | 180 (148–204) | 40–298 |
| Total, cholesterol after (mg/dL) | 157 (120–200) | 180–450 |
| | **(%)** | |
| Survival | 617 (68) | |
| Smoking | 520 (57) | |
| Ethylisms | 508 (56) | |
| Hypertension | 370 (41) | |
| *Diabetes mellitus* | 222 (24) | |
| Infections | 259 (29) | |
| Kidney disease | 80 (8) | |

References: Prostate-specific antigen (PSA) 3 ng/mL; Hemoglobin male 13.8–17.2 g/dL; Leucocyte/Lymphocyte (L/L) index, Creatinine 0.74–1.35 mg/dL; total cholesterol 125–200 mg/dL. Median. (*) = statistical difference between baseline and end of treatment $p < 0.05$ Student *t*-test.

### 3.2. Gleason Grade and PSA

Figure 1A, reveals the distribution percentage of the Gleason grade in the population studied, that grades 1 and 2 represent 50% of patients, these having a greater possibility of response to treatment with a better prognosis. Meanwhile, with higher Gleason grades, the prognosis is less favorable for survival. The Gleason grade entertains a direct relationship with the initial PSA levels as shown in Figure 1B. The majority of patients definitely had elevated PSA values.

### 3.3. Pharmacological Response to Prostate Cancer

The PSA concentration revealed a significant difference between the groups, being highest in the TRPC group with a median of 85 ng/mL (Table 2); group averages were 5.3, 10.6 and 28 times, respectively, above the reference value. The clinical differences observed were in hemoglobin (Hb) and the leukocyte/lymphocyte (L/L) ratio between baseline and end of treatment for patients with TRPC. These effects form part of ADT and possibly cancer progression. The L/L ratio exhibited greater changes in patients with TSPC and in patients with TRPC. Thrombocytopenia and leukopenia occurred in all patients with

no difference between groups; however, patients with TRPC showed an increase in the INR; this latter could be related to metastases (78%) and a higher biochemical recurrence compared to patients with TSPC (23%). Differences among patients in the therapy-sensitive group included 51% of the treated population; however, recurrence in 42% was high.

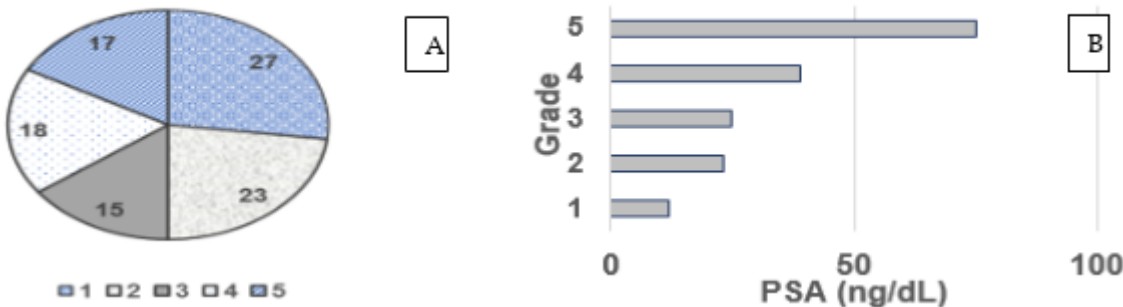

**Figure 1.** (**A**) Percentage distribution of prostate cancer cases by histological grade. (**B**) Concentration of prostate-specific antigen (PSA) vs. Gleason grade of prostate cancer.

**Table 2.** Comparison of the groups of patients regarding their response to prostate cancer treatment by ADT.

| Clinical Parameters | Hormonal Response (*n* = 466) 51% (TSPC) | Treatment Discontinuation (*n* = 60) 7% (AT) | Biochemical Recurrence (*n* = 384) 42% (TRPC) |
|---|---|---|---|
| PSA (ng/mL) | 16 ** | 32 ** | 85 ** |
| Hemoglobin baseline (g/dL) | 14.5 | 14.3 * | 13.2 * |
| Hemoglobin after (g/dL) | 13.0 | 14.0 | 11.0 |
| L/L index baseline | 4.84 * | 5.00 | 5.06 * |
| L/L index after | 6.73 * | 4.21 * | 6.89 * |
| Platelets (cells/µL) Baseline min–max | 229 ± 122 [10–662] | 236 ± 137 [92–618] | 244 ± 120 * [9–719] |
| Platelets (cells/µL) after min–max | 221 ± 125 [6–606] | 206 ± 120 [9–719] | 221 ± 137 [1–681] |
| INR (seconds) Baseline min–max | 1.01 ± 0.51 [0.75–2.26] | 1.10 ± 0.55 [0.86–2.01] | 1.11 ± 0.77 [0.76–10.93] |
| INR (seconds) after min–max | 1.00 ± 0.49 [0.76–1.70] | 1.12 ± 0.34 [0.86–2.01] | 1.17 ± 0.64 * [0.76–4.19] |
| Creatinine baseline (mg/dL) | 1.20 ± 0.9 [0.5–8.4] | 0.90 ± 0.5 [0.5–1.8] | 1.40 ± 2.7 [0.3–4.6] |
| Creatinine after (mg/dL) | 1.40 ± 1.5 [0.5–13] | 1.10 ± 0.5 [0.4–3.4] | 1.40 ± 1 [0.2–12.5] |
| Total cholesterol baseline (mg/dL) | 174 | 150 | 184 |
| Total Cholesterol (mg/dL) | 162 | 182 | 156 |
| Metastasis frequency *n* * (%) | 108 (23) | 22 (37) | 301 (78) |

Prostate-specific antigen (PSA) 3 ng/mL; Hemoglobin male 13.8–17.2 g/dL; Creatinine 0.74–1.35 mg/dL; total cholesterol 125–200 mg/dL. Platelets (147–384 × $10^3$ cells/µL), INR (2–4 s). Median: Before diagnostic analyses. After: the end treatment. ANOVA $p < 0.05$. (*) = statistical difference between baseline and end of treatment $p < 0.05$. (**) = statistical difference between baseline and end of treatment $p < 0.001$.

### 3.4. Hemoglobin Ratio and Treatment

Regardless of pharmacological and/or surgical treatment, patients showed changes in hematologic response with changes in hemoglobin. Once anemia was identified in patients

with prostate cancer, the effect was assessed at diagnosis and at the end of treatment (Figure 2); all patients with anemia were observed with an average reduction of 1 g/dL and 2 g/dL, which were statistically significant at the end of the treatment independently of the procedures, a condition that exerts an impact on clinical deterioration and the stability of the patient in terms of their QoL. Anemia is observed after 10 months of treatment and is notable in the highest-grade cases in the Gleason classification.

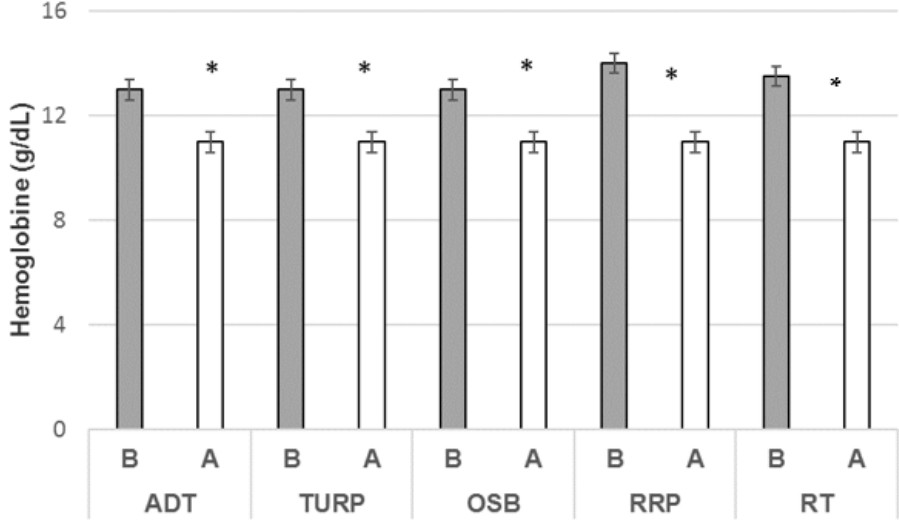

**Figure 2.** Effect of treatment on prostate cancer and hemoglobin concentration at diagnosis and the end of treatment. B = baseline; A = after. ANOVA Tukey test, $p < 0.05$. (*) = statistical differences. ADT (Androgen Deprivation Therapy); TURP (Transurethral Resection of the Prostate); OSB (Bilateral Scrotal Orchiectomy); RRP (Radical Retropubic Prostatectomy); RT (Radiotherapy).

### 3.5. Leukocyte/Lymphocyte Ratio (L/L)

Contrary to the decrease in hemoglobin in any of the treatments, the response of white cells indicated as the L/L ratio (Figure 3), independently of the treatment increases by at least 2 units. This behavior could reflect inflammatory processes, immunological response to treatment(s), infections and/or active disease.

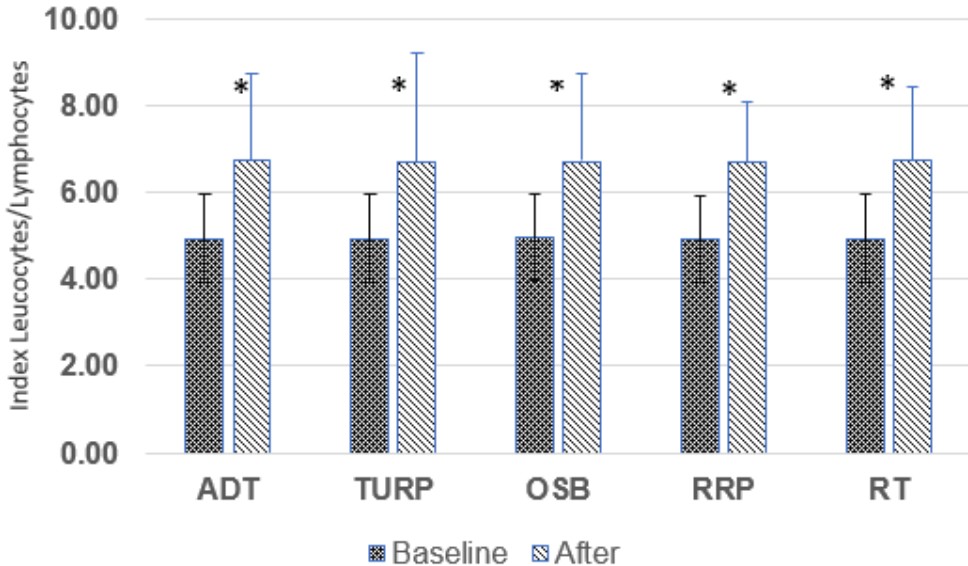

**Figure 3.** Effect of treatment on prostate cancer with respect to the Leukocyte/Lymphocyte index. ANOVA Tukey test, $p < 0.05$. (*) statistical differences. ADT (Androgen Deprivation Therapy); TURP (Transurethral Resection of the Prostate); OSB (Bilateral Scrotal Orchiectomy); RRP (Radical Retropubic Prostatectomy); RT (Radiotherapy).

### 3.6. Platelets and INR Ratio

The platelet count in the population with prostate cancer was observed within the reference values; however, when graphing by groups (TSCP, treatment abandonment (TA) and TRPC) the changes indicated in Figure 4 showed that the lowest number was observed for groups 1 and 2 at the beginning and the end, while group 3 (TRPC) demonstrated significant differences at the beginning and the end of treatment, as well as statistical differences. The platelet counts correlated with the INR, this indicated in the average of the population (GB and GA) and groups 1, 2 and 3 at the beginning and the end of treatment. The increase in INR occurs as a consequence of a lower platelet count; it is suggested that platelets could participate in processes of recurrence and metastasis in group 3.

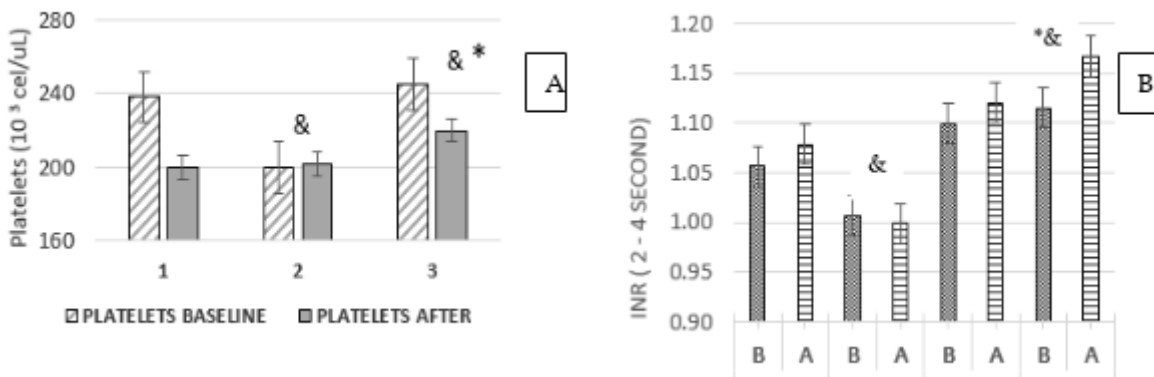

**Figure 4.** (**A**) Average platelet cells before and after treatment by group (1 = TSPC, 2 = abandonment and 3 = TRPC). ANOVA Tukey test, (&) = differences between groups; (*) difference before and at end of treatment. (**B**) INR in patients with prostate cancer assessing coagulation times. Group (1 = TSPC, 2 = abandonment and 3 = TRPC). B = before, A = after treatment. ANOVA Tukey test, (&) = differences between groups, (*) difference before and at end of treatment.

### 3.7. Platelets and Treatment

Surgical procedures and ADT were analyzed to assess their effect on platelet count and INR. Regarding the platelet level before and after the treatments (Figure 5), it was observed that the different treatments applied to patients with prostate cancer exert an impact with a decrease in circulating platelets with respect to the initial baseline, with these being the highest in OSB. The lower number of platelets correlated with longer clotting time as observed in OSB. Patients with RT showed a decrease in platelets and coagulation, the latter explained as a secondary effect due to radiation, and even due to the effects of rectal bleeding after treatment as the main symptom of radiation.

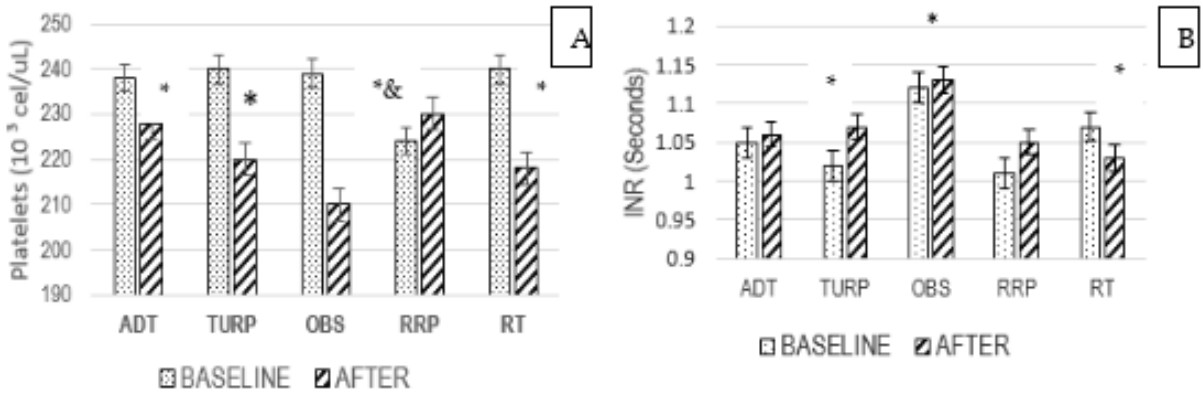

**Figure 5.** (**A**) Effect of therapy on platelet count before and after treatment in patients with prostate cancer. ANOVA Tukey test, $p < 0.05$. (*) significant differences before and after treatment. ADT (Androgen Deprivation Therapy); TURP (Transurethral Resection of the Prostate); OSB (Bilateral Scrotal

Orchiectomy); RRP (Radical Retropubic Prostatectomy); RT (Radiotherapy). $p < 0.05$ (&) significant differences between RRP vs RT, OBS. (**B**) Effect of therapy on INR before and after in patients with prostate cancer. ANOVA Tukey test, $p < 0.05$. (*) significant differences before and after treatment. ADT (Androgen Deprivation Therapy); TURP (Transurethral Resection of the Prostate); OSB (Bilateral Scrotal Orchiectomy); RRP (Radical Retropubic Prostatectomy); RT (Radiotherapy).

### 3.8. Age and Hematological Response

Age as a risk factor in the development of prostate cancer was contrasted with the hematological response (Table 3) as well as with the group in response to ADT treatment. This showed a decrease in hemoglobin, an increase in the L/L ratio, in platelets and in the infection count. These factors had statistical significance in the age range above 51 years for groups 1 and 3. Age over 51 years had hemoglobin depletion, a high response in the L/L ratio, and increased UTI (groups 1 and 3). The platelets exhibited changes in the average at the end of treatment, with maximal values in the biochemical recurrence group.

**Table 3.** Hematological data on red cells, leukocyte index, platelets, and the response to infections with respect to age range and group of patients.

| Group | Age Range | Hb (mg/dL) | | L/L Index | | Platelets | | Infections | |
|---|---|---|---|---|---|---|---|---|---|
| | | Baseline | After | Baseline | After | Baseline | After | Baseline | After |
| 1 | <50 | 14 ± 6 [9–16] | 12 ± 7 [6–17] | 4.5 ± 2 [2.4–8] | 6.9 ± 2.8 * [3.8–13] | 237 ± 107 [119–323] | 235 ± 112 [131–360] | 1 | 2 |
| | 51–70 | 15 ± 7 [6–19] | 13 ± 7 * [8–17] | 4.7 ± 23 [1.3–17] | 6.75 ± 6.0 * [2.0–33] | 230 ± 11 [36–662] | 223 ± 127 [24–606] | 29 | 35 |
| | >71 | 14 ± 7 [6–20] | 13 ± 7 * [6–18] | 5.0 ± 4 [2.0–20] | 7.1 ± 7 * [2.1–50] | 230 ± 127 [10–500] | 220 ± 122 [6–495] | 42 | 18 |
| 2 | 51–70 | 13 ± 7 [4–17] | 14 ± 6 * [9–20] | 4.0 ± 2 [2.0–20] | 4.0 ± 2.1 [2.7–10] | 233 ± 149 [92–618] | 212 ± 97 * [79–279] | 3 | 2 |
| | >71 | 14 ± 7 [8–16] | 13 ± 5 [6–14] | 4.5 ± 2.5 [2.0–14] | 3.9 ± 2.3 [1.6–10] | 241 ± 150 [143–445] | 186 ± 111 [123–237] | 3 | 1 |
| 3 | <50 | 12 ± 5 [9–15] | 8 ± 4 ** [6–11] | 5.3 ± 4.9 [3.1–8.0] | 6.7 ± 5.0 * [5.0–33] | 320 ± 150 [226–526] | 243 ± 200 [123–681] | 3 | 3 |
| | 51–70 | 13 ± 4 [5–20] | 11 ± 5 ** [4–17] | 5.2 ± 4 [1.7–20] | 7.1 ± 7.0 * [0.9–33] | 249 ± 110 [30–616] | 236 ± 144 * [1–680] | 34 | 36 |
| | >71 | 13 ± 6 [4–16] | 11 ± 5 ** [3–17] | 5.0 ± 4.5 [2–0–33] | 6.7 ± 7.0 * [1.4–50] | 238 ± 120 [9–719] | 209 ± 123 * [27–520] | 24 | 23 |

Groups: (1 = TSPC, 2 = dropout and 3 = TRPC). Hemoglobin means 13.8–17.2 g/dL; Platelets (147–384 × $10^3$ cells/µL), INR (2–4 s). Mean ± Standard deviation (SD). Baseline: diagnostic analyses. After: the end treatment. ANOVA $p < 0.05$ (*) significant differences before and after treatment. $p < 0.001$ (**) significant differences among groups.

In Tables 3 and 4, at the end of ADT treatment, it is shown that the number of patients under 50 years of age was only 2%, this group showed that the increase in Gleason grade 3 is aggressive with low survival. The age ranges of 51–70 years and over 70 years include the majority of patients in Gleason grades 1 and 2, a circumstance that favors response to treatment and possibly better survival. Survival at any age decreases with a higher Gleason grade, although in comparison, patients <50 years of age were those who had low survival which may be related to low control in androgen deprivation in young patients.

**Table 4.** Frequency distribution by Gleason grade and live patients by age group at the end of ADT treatment.

| Gleason Grade | Age Range (Years) | | | | | | | | |
| --- | --- | --- | --- | --- | --- | --- | --- | --- | --- |
| | <50 | | | 51–70 | | | >70 | | |
| | Patients | Survival | Survival (%) | Patients | Survival | Survival (%) | Patients | Survival | Survival (%) |
| 1 | 2 | 2 | 100 | 132 | 117 | 89 | 98 | 83 | 85 |
| 2 | 3 | 3 | 100 | 119 | 84 | 70 | 80 | 49 | 61 |
| 3 | 1 | 0 | 0 | 72 | 53 | 74 * | 55 | 42 | 76 * |
| 4 | 10 | 4 | 40 | 78 | 58 | 81 * | 63 | 40 | 64 * |
| 5 | 1 | 0 | 0 | 76 | 28 | 37 * | 67 | 35 | 52 * |

(*) = significant II between age range > 50 and grade cancer respective to live.

## 4. Discussion

In prostate cancer, patients are treated for androgen deprivation which limits the production of testosterone. There are effects on the 'patient's hematopoietic system with a reduction in hemoglobin of between 1 and 2 g/dL with anemia in all patients. ADT possesses a broad spectrum that will delimit testosterone as a control element; however, the approximate time ranges from 2 to 4 years and the disease may relapse with increases in PSA, as observed for this population with 42% recurrence.

In these patients, their hematological system reflects changes that can be a guide to the control condition and/or to the progression of cancer. Platelets in patients with resistance to treatment could not only have premature loss or early cell death, but also patients in the palliative stage who present thrombocytopenia, or even an over-activation that leads to thrombosis, in both extremes, a condition of disease progression. In platelets from patients with prostate cancer [16,33] there is the attribution of their participation in metastasis, in addition to a protective role on circulating tumor cells. When circulating tumor cells come into contact with platelets, they release TGF-β and molecules such as ATP, VEGF and PDGF, a condition that limits the immune response to attacking tumor cells by evading and controlling the immunological system even when high L/L ratios are observed. Although the L/L ratio increases, there is not necessarily a limitation in tumor growth, especially in cases of recurrence. This condition may explain how patients with resistance to ADT have higher platelet counts compared whit patients with sensivity to treatment [34].

Thrombocytopenia and higher INR in patients with prostate cancer and resistance to ADT treatment could be explained as a high sensitivity to platelet activation perhaps in response to signals from tumor cells. In 2020 Rudzinski et al. [16] demonstrated that the platelets of patients with TRPC and in treatment with ADT induce the target genes of the androgen receptors, with an increase in the proliferation of tumor cells via TGF-β, demonstrating that the increase in testosterone concentration occurs via an intact biosynthetic pathway identified in platelets that leads to the initiation of the cholesterol precursor pathway. This may explain how platelets from patients with resistance to ADT have high testosterone levels and even tumor proliferation with metastases as observed in a study group [35,36].

The ADT in more than 89% of patients with prostate cancer evaluated that the number of cases with recurrence (42%) and resistance to treatment is high [37]. These conditions suggest that alternate processes in the production of androgens involve 5-dyhydrotestosterone [11,19,38] in the prostate reactivating the production of testosterone from the cholesterol biosynthetic pathway when receptors are activated in platelets, hence the importance of these cells in cancer that evades initial ADT treatment. The INR data as a measure of coagulation in these patients was sensitive with an inverse correlation with respect to the platelet count, that is, low platelet count and high clotting time, observed in the treatments with OSB and in group 3 of biochemical recurrence.

Prostate cancer resistant to ADT treatment loses control when alternative metabolic pathways are activated and are mediated by cells such as platelets. Thrombocytopenia and anemia observed in patients with resistance to treatment reached 11% and 2.2%, respectively. In the remainder of the patients, although the platelet count fell within the reference limits, its participation in the resistance to treatment cannot be ruled out, as has been demonstrated in cell models [30,33].

Patients manifest a reduction in hemoglobin, but at the same time the white cells increase the L/L ratio, that is, the hematological condition is not only a function of the incidence of infections but also of cancer control, modifying the role of at least platelets and coagulation times (INR); both responses appear to favor tumor cells in the metastases process in at least 50% of the cases studied, a process with a high incidence in bone tissue, organs (liver, kidney, bladder, colon and lung), followed by local progression and some cases of the central nervous system (CNS).

Anemia by loss of hemoglobin is also reported in heavy metal poisoning and has been shown to inhibit the enzymes of this biosynthesis pathway [39,40]; although, in the case of prostate cancer treatment, ADT could also be interfering with the synthesis of hemoglobin. It is also important to take into consideration that the accumulation of substrates such as aminolaevulinic acid leads to the formation of enolized and oxidized compounds (4,5-dioxovaleric), with an increase in reactive oxygen species [41], and an increase in peroxides and in superoxide radicals that affect DNA bases. The loss of blood cells could be due to increased eryptosis and apoptosis as has been suggested [30,35].

The increase in the L/L ratio in patients with prostate cancer appears to be a response to UTI; without being altered by age, this would indicate that the immune system in the white cells is not altered, or even, that the immune system possesses high activity in response to co-morbidity factors such as obesity and overweight being present in 50% of the study population as a chronic pro-inflammatory state with a high L/L index independently of the treatments taking place in the patients. As a whole, it can be understood that the systemic imbalance of prostate cancer adds to the chronic inflammatory process due to obesity and hypertension with a direct impact on endothelial tissue and constant platelet activation, as has been suggested [15,16]. Obesity as a comorbidity presents chronic inflammation, and changes in the type of inflammatory macrophages M2, by itself do not generate evident hematological changes. However, it is possible that adipose tissue has implications in patients with prostate cancer and may limit the effectiveness of treatment in cases of obesity.

As shown, prostate cancer has a direct relationship with the Gleason grade and the PSA concentration. Hormone inhibition therapy does not work in 50% of patients with a Gleason grade higher than 3, that is, hormonal inhibition works only in low Gleason grades, where 51% of the population studied had a survival greater than 70% after 5 years [27,42]. Young patients with prostate cancer are more likely to die: it is assumed that they have a failure to respond to treatment in hormonal deprivation, or that they have a higher number of androgen receptors compared to adults over 60 years of age.

Androgen deprivation in prostate cancer generates a different immune response among erythrocytes, lymphocytes and platelets. Patients with this diagnosis confirm anemia during treatment. At the same time, there is an important response of lymphocytes (L/L ratio) and platelets, especially in patients with recurrence (47–50%). In patients with thrombocytopenia, the preponderant condition in the patients is frequent fatigue and walking limitations with high clinical deterioration.

## 5. Conclusions

Patients with prostate cancer exhibit a direct correlation between the Gleason grade and a high PSA concentration, which is related to the risk and to recurrence. ADT exerts changes on cells of the hematologic system with anemia, L/L activation and thrombocytopenia. Hormonal deprivation affects the patients' entire physiological status. The evident inflammatory response at the end of treatment (L/L) could be related to the chronic inflam-

matory response due to obesity and hypertension plus ADT in these patients. Androgen inhibition and its effects on the hematological system affect the physical condition of all of the patients due to anemia and thrombocytopenia. The INR in these patients could represent an indicator of the response of the platelet count for prognostic purposes of the patient for treatment adjustments.

**Author Contributions:** All authors contributed to the study conception and design. Material preparation, data collection and analysis were performed by S.C.-A., M.M.-V., M.A.R.-R., E.M.-C., M.A.B.-S. and F.F.-S. The first draft of the manuscript was written by S.C.-A., M.M.-V. and F.F.-S. and all authors commented on previous versions of the manuscript. All authors have read and agreed to the published version of the manuscript.

**Funding:** This research received no external funding.

**Institutional Review Board Statement:** The study was conducted according to the guidelines of the Declaration of Helsinki and was approved by the Bajio Regional High Specialty Hospital in León, Guanajuato, Mexico through Research (CI/HRAEB/2021/019 and Ethics (CNBCE-11-CEI/004-20170731; CEI-022-2021). The study used a retrospective database, The follow-up of the patients and their treatment supplies correspond to the health care schemes of the population under the responsibility of the Mexican Federal Health Secretariat, which includes High Specialty Hospitals distributed throughout the Mexican Republic.

**Informed Consent Statement:** Informed consent was obtained from all subjects involved in the study when they were diagnosed, and gave consent to use their data in subsequent studies.

**Data Availability Statement:** The data were captured from the KlinicRM database of the Bajio Regional High Specialty Hospital, after authorization by the research and research ethics committees, but the information is not public due to ethical safekeeping of patient information.

**Acknowledgments:** Bajio Regional High Specialty Hospital for financial support for the publication, and to the Department of Planning, Teaching and Medical Research Esperanza García-Moreno.

**Conflicts of Interest:** The authors declare no conflict of interest.

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
