# Peer review of "Prostate Cancer, Treatment and Response of the Hematological System in Mexican Population"

_2673-8937, doi:10.3390/ijtm3030020_

Round 1

Reviewer 1 Report

In this study, the authors have evaluated how standard treatment affects the hematological system of prostate cancer patients. 910 patients treated with ADT presented with anemia regardless of surgery or radiotherapy. The hematological response due to the leukocyte/lymphocyte index is increased at the end of treatment, possibly due to inflammation. Patients with biochemical relapse exhibit a higher platelet count, suggesting that these cells could participate in the recurrence process and in metastasis in these patients. The coagulation index could be an indicator of the platelet response to be considered during the treatment and monitoring of patients.

The study is informative and relevant, as there is concern regarding the increased burden of prostate cancer in the Mexican population. The manuscript would greatly benefit from dissecting the population based on different criteria and patient outcomes. These correlations would increase disease awareness and diagnostic practices.

Below I provide comments to improve the paper.

1.    The study would benefit from a further breakdown of the Mexican population according to geographical location or racial background. Ethnic heterogeneity impacts patient outcome, and it might shed light on different aspects of this study.

  1. The author should include images with better resolutions.
  2. The manuscript should be evaluated for typos and spelling errors.
  3. How early were patients diagnosed with anemia during their treatment period?

The manuscript should be evaluated for typos and spelling errors.

Author Response

The response to your comments is added and incorporated into the document. Thank you.

Reviewer 2 Report

This article describes the research on the treatment effects on the response of prostate cancer patients' hematological system in the Mexican Population. The authors classified the patient group according to their Gleason grade, age, and sensitivity to the treatment. The results showed the difference between baseline and after the treatment, and it will be helpful for follow-up the prognosis. It would be better if the author included the results with Gleason's grade. The treatment results should differ from the cancer progression, depending on their Gleason grade. How effective the treatment is and how they respond in the hematological system would be helpful. Also, it would be beneficial if the author investigated the effects of obesity with ADT therapy.

Throughout the manuscript, it has grammatical errors. I recommend publishing this manuscript in the International Journal of Translational Medicine with the following minor revision and correcting all English errors. Abstract must be Abstract. The authors included “purpose,” “results,” and “conclusions.” These must not be labeled in abstract.

·      In the text body, one-space and two-space are mixed. Please make them consistently.

·      Figure 1-B. B is overlapped with grade 5.

·      Table 2 has * and ** but there is no explanation on the footnote.

·      Figure 3, error bar cannot be shown properly in “After” data. Also, it is hard to see in other figures, too. I recommend change the shades.

·      Table 4 has * but there is no explanation.

·      If Table 4 showed the comparison of after ADT treatment, please mention in the manuscript and the Table caption.

·      Figure 4-B. asterisk * must be right upper corner on &. Also, error bar disappeared on the 2nd B (below &)

·      In Table 4, the authors use the word “Frequency” but this is shown the number of patients. Also, they use “live patients” but generally “survival” is used as a medical term.

In addition to the comments above, there are some grammatical and punctuation errors throughout the manuscript. I recommend the authors to proofread it carefully.

Line 350 the preponderant condition in the patient is frequent fatigue, limitation in walking with high clinical deterioration.--> the preponderant conditions in the patient is  are frequent fatigue and  limitation in walking with high clinical deterioration.

Line 356 patient’-s  --> patients’

In author contributions, they used “y” (Spanish), not “and.”

Author Response

(The authors gave the same response as above.)
